# A Review on the Role of Bicarbonate and Proton Transporters during Sperm Capacitation in Mammals

**DOI:** 10.3390/ijms23116333

**Published:** 2022-06-06

**Authors:** Ariadna Delgado-Bermúdez, Marc Yeste, Sergi Bonet, Elisabeth Pinart

**Affiliations:** 1Biotechnology of Animal and Human Reproduction (TechnoSperm), Institute of Food and Agricultural Technology, University of Girona, ES-17003 Girona, Spain; ariadna.delgado@udg.edu (A.D.-B.); marc.yeste@udg.edu (M.Y.); sergi.bonet@udg.edu (S.B.); 2Unit of Cell Biology, Department of Biology, Faculty of Sciences, University of Girona, ES-17003 Girona, Spain; 3Catalan Institution for Research and Advanced Studies (ICREA), ES-08010 Barcelona, Spain

**Keywords:** HVCN1 channels, SLC4 channels, SLC26 channels, NHE, sperm capacitation, mammals

## Abstract

Alkalinization of sperm cytosol is essential for plasma membrane hyperpolarization, hyperactivation of motility, and acrosomal exocytosis during sperm capacitation in mammals. The plasma membrane of sperm cells contains different ion channels implicated in the increase of internal pH (pH_i_) by favoring either bicarbonate entrance or proton efflux. Bicarbonate transporters belong to the solute carrier families 4 (SLC4) and 26 (SLC26) and are currently grouped into Na^+^/HCO_3_^−^ transporters and Cl^−^/HCO_3_^−^ exchangers. Na^+^/HCO_3_^−^ transporters are reported to be essential for the initial and fast entrance of HCO_3_^−^ that triggers sperm capacitation, whereas Cl^−^/HCO_3_^−^ exchangers are responsible for the sustained HCO_3_^−^ entrance which orchestrates the sequence of changes associated with sperm capacitation. Proton efflux is required for the fast alkalinization of capacitated sperm cells and the activation of pH-dependent proteins; according to the species, this transport can be mediated by Na^+^/H^+^ exchangers (NHE) belonging to the SLC9 family and/or voltage-gated proton channels (HVCN1). Herein, we discuss the involvement of each of these channels in sperm capacitation and the acrosome reaction.

## 1. Introduction

Sperm capacitation occurs in the female reproductive tract and is essential for sperm to trigger acrosomal exocytosis and further oocyte fertilization. The biochemical and biophysical changes associated with sperm capacitation have been extensively reported, and they include changes in ion concentration and internal pH (pH_i_), plasma membrane properties, motion features, cell metabolism, and protein phosphorylation patterns [1,2,3,4]. It is widely accepted that ion flux is essential for sperm capacitation, and that small disturbances in the activity of ion channels may result in impaired fertility [1,5,6,7,8]. The most relevant ions inducing sperm capacitation are bicarbonate (HCO_3_^−^) and calcium (Ca^2+^), which act as second messengers triggering both sperm hypermotility and acrosomal reaction [1,9]. In effect, intracellular HCO_3_^−^ and Ca^2+^ bind to the soluble adenylate cyclase (sAC), thereby increasing adenosine monophosphate levels (cAMP), which in turn activate the protein kinase A (PKA) [10]. The PKA is responsible for the phosphorylation cascades activating different ion channels implicated in cytoplasm alkalinization [11], which leads to increased membrane fluidity following cholesterol efflux [12,13,14]; membrane hyperpolarization [15,16]; and phosphorylation of several flagellar proteins including axonemal and peri-axonemal, and others involved in metabolism [1]. Inner Ca^2+^ also activates phospholipase C (PLC), generating 1,4,5-triphosphate (IP3) and diacylglycerol (DAG), that are second messengers of the IP3 receptor and protein kinase C (PKC), both involved in the regulation of the acrosomal reaction [9,17]. Finally, Ca^2+^ also binds to the calmodulin present in the sperm head and flagellum, thus activating other additional phosphorylation cascades regulated by calmodulin kinase that are also essential for both hyperactivation of sperm movement and acrosomal exocytosis [1,16].

Bicarbonate influx is the first event associated with sperm capacitation and is essential for switching on the PKA pathway and pH_i_ alkalinization, and further membrane hyperpolarization [1,9,16]. For this reason, the sperm plasmalemma contains different types of bicarbonate transporters differing in solute affinity, stoichiometry, and regulation mechanisms; these channels are currently grouped into Na^+^/HCO_3_^−^ transporters and Cl^−^/HCO_3_^−^ exchangers and belong to the superfamily of solute carrier transporters (SLC) [1,16,18,19]. Yet, it should be borne in mind that intracellular HCO_3_^−^ levels also rely upon different carbonic anhydrase isoforms present in both plasma membrane and cytoplasm [20]. On the other hand, not only does alkalinization of sperm cytosol depend on bicarbonate transport but also on the involvement of different proton (H^+^) exchangers and transporters, mainly voltage-gated proton channels, such as HVCN1 [3,21,22], monocarboxylate transporters (MCTs) [19], and Na^+^-H^+^ exchangers (NHEs) [8,23,24,25]. In fact, the increase of pH_i_ is known to be essential for the activation of several pH-dependent channels that play a vital function for plasma membrane hyperpolarization and acrosomal reactions [1].

Calcium influx is mainly driven by sperm-specific pH sensitive voltage-gated Ca^2+^ channels (CatSper) [26,27]. Due to their biological relevance for male fertility, the structure, localization, types, expression, and regulation mechanisms of CatSper channels have been extensively investigated [28,29,30,31,32,33,34,35]. It is, nevertheless, worth noting that the sperm plasma membrane also has other channels, such as Ca^2+^ transporters, which participate in the calcium rise during sperm capacitation (e.g., Ca^2+^-ATPases; [36,37]; transient-receptor potential (TRP) channels [38,39]; Na^+^/Ca^2+^ exchangers (NCX) [37,40]; voltage-gated Ca^2+^ channels (Ca_V_) [41]; and cyclic nucleotide-gated (CNG) channels [42]).

Apart from the transport of Ca^2+^, HCO_3_^−^, and H^+^, sperm capacitation also requires a change in the flux of other ions across the plasma membrane; among them, one should highlight potassium (K^+^), chloride (Cl^−^), and sodium (Na^+^), as they are needed to induce membrane hyperpolarization (Em), sustain the Ca^2+^ influx, and trigger acrosomal exocytosis [1,16,20]. Despite their physiological relevance, little data exist on their conductance through the plasma membrane in mammal sperm, with most studies having been performed in humans and rodents. K^+^ transporters implicated in sperm capacitation are SLO channels [2,43] and voltage-gated channels (KCNQ or K_v_7) [44,45], whereas Cl^−^ transport is mediated by Ca^2+^-activated Cl-channels (CaCCs) [46], cystic fibrosis conductance channels (CFRT) [16,47], Na^+^/K^+^/Cl^−^ cotransporter (NKCC) [16,48,49], and voltage-dependent anion channels (VDAC) [50]. Finally, Na^+^ transport is driven by thermo-sensitive transient-receptor potential vanilloid channels (TRPVs) [51], epithelial sodium channels (ENaC) [16], and voltage-gated sodium channels (Na_V_) [52].

The present review gathers the latest knowledge on the structure, localization, physiological role, and regulation mechanisms of ion transporters implicated in cell alkalinization during sperm capacitation in mammals, including bicarbonate transporters (Na^+^/HCO_3_^−^ cotransporters and Cl^−^/HCO_3_^−^ exchangers) and proton transporters (NHE and HVCN1 channels).

## 2. Bicarbonate Transporters

Bicarbonate transporters belong to the families of solute carriers 4 (SLC4) and 26 (SLC26) [9,19]. All members of these two families are secondary ion transporters, i.e., ion-coupled transporters that require the electrochemical gradient of an ion to transport the other one against its gradient [53]. Each bicarbonate transporter (i.e., SLC4 or SLC26) is encoded by a specific gene; despite sharing a similar structure, they differ in ion affinity, transport function, physiological properties, and interaction with other proteins [19].

### 2.1. The SLC4 Family

The SLC4 family is formed by ten different members, nine HCO_3_^−^ transporters and one borate (B(OH)_4_^−^) transporter. Bicarbonate transporters are currently grouped into Na^+^/HCO_3_^−^ cotransporters and Cl^−^/HCO_3_^−^ antiporters [16,54,55]. To date, four Na^+^/HCO_3_^−^ cotransporters (NBCs) have been identified: SLC4A4 or NBCe1 and SLC4A5 or NBCe2 are electrogenic (1Na^+^:2HCO_3_^−^), whereas SLC4A7 or NBCn1 and SLC4A10 or NBCn2 are electroneutral (1Na^+^:1HCO_3_^−^). Cl^−^/HCO_3_^−^ exchangers can be either Na^+^-coupled (SLC4A8 or NDCBE and SLC4A9 or AE4) or Na^+^-independent (SLC4A1, SLC4A2, and SLC4A3) [54,55]. SLC4A1, SLC4A2, and SLC4A3, also known as anion exchangers (AE) AE1, AE2, and AE3, respectively, are electroneutral antiporters widely expressed in several tissues, although each gene produces tissue-specific variants by alternative splicing [19,54,55]. In general, Cl^−^/HCO_3_^−^ exchangers transport bicarbonate either into or out of the cells, so they can either alkalinize or acidify the cytoplasm [54]. Interestingly, the C-terminal domain of Cl^−^/HCO_3_^−^ exchangers interacts with carbonic anhydrases, which catalyze the reaction CO_2_ + H_2_O ⇄ HCO_3_^−^ + H^+^; such a physical and functional relationship modulates both substrate transport and anhydrase kinetics [56] and it has an essential role in regulating HCO_3_^−^ homeostasis in sperm cells [16]. In effect, in both mice [57,58] and humans [59], the alteration in anhydrase activity results in reduced sperm motility. Finally, there are discrepancies about the type of solutes transported by SLC4A11 or BTR1, as it could act as either an electrogenic Na^+^/B(OH)_4_^−^ cotransporter or an electrogenic H^+^ transporter activated by NH_3_ and alkaline pH [60].

SLC4 transporters are usually present as dimers, but they can also form tetramers. Each subunit has a variable amino acid (aa) length (from 850 to 1250 amino acids, aa) and molecular weight (from 90 to 200 kDa), which depends on post-translational glycosylation (Figure 1) [61]. Each polypeptide contains three major domains (reviewed in [61]): (a) the large N-terminus domain with 300 to 700 aa, which modulates ion transport and interacts with multiple cytoskeletal proteins; (b) the transmembrane domain, with 500 aa forming 14 spans, which is implicated in ion translocation across the plasma membrane; and (c) the short C-terminus domain, essential for SLC4 trafficking and surface expression.

Some functional approaches based on current voltage measurement and pharmacological inhibition indicate the presence of NBC channels in the plasma membrane of human and rodent sperm. These channels, which seem to be responsible for pH_i_ homeostasis after ejaculation [11], are involved in the initial, fast bicarbonate entrance that triggers sperm capacitation by activating the cAMP/PKA pathway [16,62] and leads to the lateral redistribution of cholesterol, an essential step for a further cholesterol depletion and increase of membrane permeability [12,14]. The seminal plasma has a high content of Na^+^ and HCO_3_^−^, which may favor the import of both ions through NBC [16]. Yet, the mechanisms via which these channels are activated immediately after ejaculation and even the types and localization of NBC implicated in this process remain unknown. In this regard, only transcripts of electroneutral SLC4A10 have been identified in human testes [63], as the protein expression in mature sperm cells has not been reported. In humans, one of the first targets of the PKA pathway is the cystic fibrosis transmembrane receptor (CFTR), which is essential for a further sustainable entrance of bicarbonate through Cl^−^/HCO_3_^−^ exchangers [16,20]. In rodents, however, not only does the sustained activity of Cl^−^/HCO_3_^−^ exchangers depend on CFTR but also on NKCC [16,48,49]. Interestingly, HCO_3_^−^ transport through Na^+^/HCO_3_^−^ transporters is independent from Cl^−^ transport [48].

NBC channels also participate in further stages of sperm capacitation. In humans, while electrogenic NBC has a minor role in pH_i_ alkalinization, it participates in motility hyperactivation and membrane hyperpolarization through the PKA pathway [11,16]. It has been suggested that electrogenic NBC maintains active CFTR channels [11], which is essential for ENaC inactivation, a relevant step for membrane hyperpolarization [20,64]. To the best of the authors’ knowledge, nonetheless, no study has previously investigated the presence and specific localization of electrogenic NBC channels (SLC4A4 and/or SLC4A5) in sperm plasma membrane.

Regarding SLC4 channels, which, as aforementioned, work as Cl^−^/HCO_3_^−^ exchangers, only Na^+^ independent SLC4A1 [1,9,16] and SLC4A2 [65] are known to be present in mature sperm from humans. SLC4A1, also known as AE1 or the band 3 channel, has a wide distribution in different tissues, including the intestine, kidney, heart, and epididymis [66,67]. It is an electroneutral exchanger (1:1) of several monovalent anions, with the most currently transported solutes being Cl^−^ and HCO_3_^−^ [9]. SLC4A1 exchangers usually form dimers or tetramers; the cytosolic domains have different binding sites that can interact with cytoskeletal proteins and enzymes [9,68], although the activity and interaction capacity rely on the phosphorylation status of specific residues [69]. The channel activity is thus regulated by phosphorylation of serine (Ser) residues at the cytoplasmic domain [70,71] (Baggio et al., 1993, 1999), whereas the interaction with other proteins and enzymes is regulated by tyrosine (Tyr) phosphorylation [68,72]. In human sperm, SLC4A1 distributes throughout the plasma membrane of both the head and flagellum and has a molecular weight of 90 kDa [9]. During sperm capacitation, SLC4A1 does not seem to have an essential role in the regulation of sperm motility or in the maintenance of the structural integrity of the fibrous sheath during the hyperactivated movement; in contrast, it appears to be essential for acrosomal reaction [9]. The sperm head has two specific protein kinases: (a) the spleen tyrosine kinase (Syk) that phosphorylates different Tyr residues implicated in the recruitment of SLC4A1 channels in specific plasma membrane regions of the acrosomal domain, and (b) the Lck/Yes novel (Lyn) tyrosine kinase that phosphorylates the Tyr residue 359 of SLC4A1 (Tyr359; [9]), as well as other head proteins [73]. Specifically, the phosphorylation of Tyr359 induces a conformational change in the N-cytoplasmic terminus that promotes the dissociation between SLC4A1 and the spectrin-actin network [74] and the depolymerization of actin [75] just before the acrosomal reaction. In idiopathic infertile patients, the alterations in Lyn activity result in defective Tyr359 phosphorylation and, therefore, impaired acrosomal reaction [73].

SLC4A2 channels are already expressed in differentiating germ cells from rodents and humans, where they could be essential for spermatid elongation [65,76]. Null *Slc4a2* mice are infertile due to few elongating spermatids and absence of sperm cells in the seminiferous tubules [76]. Other NBC channels could also be essential for spermatogenesis as suggested by the presence of *SLC4A4*, *SLC4A8*, and *SLC4A10* transcripts in differentiating germ cells from humans [63]; nevertheless, the presence of these channels in the plasma membrane of mature sperm has not been reported. In mature sperm from mice and humans, SLC4A2 is located in the equatorial region [65]; however, its specific role in sperm capacitation has not been described.

### 2.2. SLC26 Family

The SLC26 family is formed by 11 electrogenic anion members, each encoded by a specific gene (*SLC26A1*–*SLC26A11*), sharing a common structure but differing in stoichiometry and solute specificity. In fact, not only do the members of this family transport Cl^−^ and HCO_3_^−^, but also sulphate (SO_4_^2−^), iodide (I^−^), formate (HCOO_3_^−^), and oxalate (C_2_O_4_^2−^) [53]. It is important to highlight that only five members are permeable to HCO_3_^−^: SLC26A3, SLC26A4, SLC26A6, SLC26A7, and SLC26A9 [19,53,54].

SLC26 channels have two different regions (Figure 2): (1) a highly conserved transmembrane region with 10–14 spans, which is implicated in anion transport, and (2) a cytoplasmic region formed by the N- and C- terminal domains that includes the sulphate transporter and anti-sigma factor antagonist domain (STAS domain) at the C-terminus, involved in the expression, activity, trafficking, and protein–protein interaction of SLC26 [1,77,78]. Moreover, several SLC26 channels also have a PDZ binding domain at their C-terminus, which mediates their clustering in the plasma membrane [53,78,79,80]. Although these channels can act as monomers, they usually associate forming either homo-/hetero-dimers or tetramers, each subunit being involved in anion translocation but interacting with each other [53,81]. In several tissues, SLC26 channels can also interact with other ion channels, such as CFTR [82] and NHE [83,84,85], through the STAS domain.

SLC26 channels are distributed throughout the body, showing an organ-specific distribution [53]; however, only SLC26A3 and SLC26A6 are reported to be present in the sperm plasma membrane [81], whereas SLC26A8 is sperm-specific [19,53,82]. In SLC26A3 and SLC26A6 channels, the HCO_3_^−^ import is coupled to Cl^−^ export, with similar permeability to both ions but different stoichiometry [19,53]. The SLC26A3 exchanger, with 1:1 or 2:1 stoichiometry, is mainly expressed in the intestine [86], and mutations of its gene lead to congenital chloride losing diarrhea (CLD) [77]. SLC26A6 exchangers are predominantly expressed in the pancreas, kidney, and intestine of humans and mice; they act as electroneutral Cl^−^/HCO_3_^−^ exchangers (1:1) in humans, and as electrogenic exchangers in mice (1:2) [86,87,88]. As far as we are aware, little data exist on the stoichiometry and anion specificity of SCL26A8; it is an electroneutral Cl^−^/SO_4_^2−^ transporter with an unknown affinity for HCO_3_^−^ [89,90].

In mice, *Slc26a6* and *Slc26a8* transcripts are already detected during spermatogenesis; testicular germ cells also contain *Slc26a2*, *Slc26a7*, and *Slc26a11* transcripts [91,92,93], despite protein expression in mature sperm not being reported. In humans, transcription of *SLC26A3* occurs throughout spermatogenesis and protein expression during epididymal maturation [1], whereas in mice both transcript and protein are detected in epididymal and mature sperm [18,94]. Finally, neither SLC26A1, SLC26A4, SLC26A5, and SLC26A9 transcripts nor the corresponding proteins have been identified in differentiating or mature sperm cells [91,92,93].

The presence of SLC26A3 in the sperm plasma membrane was first described by Chen et al. [95] in guinea pigs, with the channel being in the acrosomal region of the sperm head. Further immunofluorescent approaches described the presence of SLC26A3 and SLC26A6 channels in the midpiece of mature sperm from mice, despite a weak staining in the sperm head being also observed [81]. The molecular weight of these channels varies between species, probably because of different post-translational modifications; therefore, SLC26A3 is around 75 kDa in guinea pig sperm [95] and 98 kDa in those from mice [81], and SLC26A6 is <95 kDa in mouse sperm [81]. During sperm capacitation, HCO_3_^−^ transport through SLC26A3 is associated with protein phosphorylation and hyperactivation of sperm motility [1,77], whereas SLC26A6 does not seem to be relevant in mouse sperm [81]. While *SLC26A3* knockout mice are subfertile and exhibit low sperm concentration and impaired sperm motility [1,53], *Slc26a6* knockout males are fertile [1,96]. In humans, mutations in the gene encoding for these two channels are associated to male subfertility due to impaired sperm concentration, motility, and morphology, and high content of Cl^−^ and H^+^ in the seminal plasma [97].

SLC26A8 or testis anion transporter 1 (TAT1) is located in the equatorial segment and Jensen’s annulus of mouse and human sperm [53,79,90,98,99]. During spermiogenesis, this channel is expressed before the assembly of septin complex in the annulus [100]; the absence or abnormal expression of SLC26A8 manifests in an impaired differentiation or even in the absence of the annulus and mitochondrial sheath, thereby resulting in severe asthenozoospermia [44,79]. During sperm capacitation, disturbances in SLC26A8 function lead to impaired motility hyperactivation and acrosomal exocytosis because of defective sAC inactivation and low cAMP levels [81,90,98].

In sperm cells, in a similar fashion to other cell types, SLC26 antiporters are physically and functionally related to CFTR channels, which are cAMP-dependent Cl^−^ channels activated by PKA phosphorylation during the first stages of sperm capacitation [16,53,81]. In mouse, guinea pig, and human sperm, CFTR channels show a similar location as SLC26 antiporters; they are all located in the equatorial region and midpiece, which favors their association through the STAS domain of SLC26 and the regulatory (R) domain of CFTR [90,101,102,103]. It is worth noting that this interaction requires the phosphorylation of both domains via PKA [81,104]. CFTR also has a PDZ domain, so the association with SLC26 results in the formation of protein complexes with two PDZ domains that can interact simultaneously with binding and scaffolding PDZ-proteins, thus forming large protein complexes associated with the plasma membrane [81,105,106,107].

In capacitated sperm, SLC26-CFTR complexes show higher transport activity than single channels due to mutual activation of their function [20,53,90,104,108], which favors the maintenance of a sustained and high bicarbonate entrance via Cl^−^/HCO_3_^−^ exchangers [16] and Cl^−^ inward through CFTR. Furthermore, Cl^−^ influx through CFTR is known to be essential for the maintenance of sAC/PKA pathways, acting together with Ca^2+^ and HCO_3_^−^ [1,48,90,101]. It is, however, worth highlighting that the activation of SLC26 does not just require the presence of CFTR but also its functional triggering [81,85]. Previous research showed that the inhibition of CFTR channels leads to defective pH_i_ alkalinization and activation of the cAMP/PKA pathway, which results in impaired motility hyperactivation and acrosome reaction [1,16,20,103]. In several tissues, CFTR also regulates the activity of other channels, such as ENaC [101], NHE [1], and aquaporins [109,110]. In human sperm, ENaC and CFTR co-localize in the midpiece of the flagellum [101,111]; remarkably, the inactivation of ENaC by activated CFTR is reported to be an essential step for membrane hyperpolarization during sperm capacitation [20,64]. Related to this, it is worth indicating that membrane hyperpolarization also requires the activation of SLO channels, which are voltage-gated channels whose activity is also regulated by phosphorylation via the PKA pathway [112,113].

In guinea pig and mouse sperm, SLC26A3-CFTR complexes are essential for the pH_i_ increase and the maintenance of high cAMP levels and protein phosphorylation that occur during sperm capacitation [1,48,81,95]. The pharmacological inhibition of SLC26A3 results in a decrease of Cl^−^ influx and partial blockage of both pH_i_ alkalinization and membrane hyperpolarization, whereas CFTR inhibition leads to an almost blocked hyperpolarization [81]. In humans, *SLC26A3* mutations are associated with CFTR inhibition and male infertility [77], whereas *Slc26a3* deletion results in an impaired sperm fertilizing capacity in mice [18]. SLC26A3-SLC26A6-CFTR complexes have also been described to be present in the midpiece of mouse sperm, participating in pH_i_ alkalinization during sperm capacitation [81].

Physical and functional interaction between SLC26A8 and CFTR channels has also been reported [82,90]. SLC26A8-CFTR complexes play a relevant role in the hyperactivation of sperm movement and acrosomal exocytosis, via the PKA-dependent downstream phosphorylation cascade [79,90,98]. In mouse sperm, not only are SLC26A8 and CFTR channels located in the annulus but also sAC [90]; this favors a local cAMP production that may be essential for the maintenance PKA pathways in the sperm flagellum. Moreover, SLC26A8 and CFTR also co-localize at the equatorial segment of the sperm head, and their interaction seems to be essential for acrosomal exocytosis [90]. In agreement with this, the acrosomal reaction is impaired in *Slc26a8* null sperm [98], as well as in CFTR blocked human and mouse sperm [101,102].

## 3. Proton Transporters

### 3.1. Na^+^-H^+^ Exchangers

Na^+^/H^+^ exchangers (NHEs), belonging to the protein family of solute carrier 9 (SLC9), participate in the increase in pH_i_ by means of the electroneutral exchange of extracellular Na^+^ for intracellular H^+^ [19]. The SLC9 family includes several transporters, each encoded by a specific gene and differing in localization, molecular and kinetic properties, drug sensitivity, and regulation. As a result, these transporters are grouped into three different subfamilies [19,114,115]: (1) NHE1–NHE9 exchangers, which can be located either in the plasma membrane (NHE1 to NHE5 or SLC9A1 to SLC9A5) or intracellularly (NHE6 to NHE9 or SLC9A6 to SLC9A9); (2) NHA1 and NHA2 antiporters (SLC9AB1 and SLC9AB2); and (3) sperm-specific NHE exchanger (sNHE, SLC9C1 or SLC9A10).

SLC9 members are found as either monomers or dimers [114,115]; despite each monomer being able to act as a single channel, the dimerization provides stability to the channel [116]. SLC9 monomers share a similar structure with two domains (Figure 3): (1) a transmembrane domain with 11–12 spans implicated in ion exchange and pH sensing, both functions being separated spatially, and (2) a C-terminus domain of variable length located in the cytoplasm and involved in the regulation of exchange activity by interacting with the short cytoplasmic N-terminus and the phospholipids of the inner leaflet of the plasma membrane [114,116,117,118,119]. The C-terminus is also a target for protein interaction and phosphorylation [118].

Few studies have focused on the expression and function of NHE in sperm cells; NHE1, NHE5, and/or sNHE exchangers are reported to be expressed in some mammal sperm cells [23,25,114], whereas the presence of NHA1 and NHA2 has only been described in mice [120]. To our knowledge, no information is available on the presence, distribution, and function of intracellular NHE channels in mature sperm cells. Only NHE8 isoforms are described in the Golgi complex and acrosomal granules of spermatids, being essential for acrosome differentiation during spermiogenesis [121]. On the other hand, most studies about NHE function are based on the specific blockage using pharmacological inhibitors; due to the lack of a specific inhibitor for each exchanger type, while these assays provide general information, they do not offer a clue on the physiological role of each NHE type. In mice, such approaches have demonstrated that NHE blockage results in impaired sperm motility [8,120]. In this species, NHE activation during sperm capacitation is functionally coupled to K^+^ and Ca^2+^ currents, mainly through SLO and CatSper channels, respectively, which are both pH_i_ sensitive [8,11]. The increase of pH_i_, therefore, is essential for membrane hyperpolarization through SLO channels [11,122,123] and sustained Ca^2+^ influx through CatSper [122,124]. In agreement with this, blockage of NHE using 5-(*N,N*-dimethyl)-amiloride (DMA) manifests in a decrease of K^+^ currents across SLO channels and of Ca^2+^ currents across CatSper recorded using the patch-clamp technique [8,114]. In pigs, NHE blocking with DMA does not affect Ca^2+^ influx measured by flow cytometry during in vitro capacitation, but it reduces the ability of capacitated sperm cells to respond to the progesterone stimulus; this drives impaired hypermotility and acrosomal exocytosis [25]. In humans, an NHE blockage does not affect the acrosomal reaction after progesterone addition in capacitated sperm [125].

NHE1 is expressed in nearly all mammalian cells [19,126], and it mainly extrudes H^+^ derived from cell metabolism [127]. In sperm cells, the localization, molecular weight, and physiological role of NHE1 channels differs with species. Indeed, whereas they are located in the midpiece of flagellum in mice [114,128], they extend along the mid- and principal pieces and the equatorial region of the head in sheep [23] and pigs [25]. The molecular weight of NHE1 in sperm is 95 kDa in rats [128], 85 kDa in sheep [23], and 75–105 kDa in pigs [25,125]. Despite this, little data on the physiological role of NHE1 in mature sperm exist; in sheep, it has been reported to be essential for activation of sperm motility upon ejaculation; paradoxically, its pharmacological blockage does not result in an impaired sperm fertilizing capacity [23]. In mice, NHE1 does not seem to participate in the activation of pH-sensitive ion channels during sperm capacitation [8], but it could exert a local control of the intracellular H^+^ concentration [128]. Furthermore, knockout mouse males lacking this exchanger are fertile [114].

The presence of sNHE has only been reported in mice and humans, differing in molecular weight and distribution. In mice, these exchangers have a molecular weight of 120 kDa and are located in the principal piece [114], whereas in humans their molecular weight varies from 120 to 135 kDa and they extend along the mid- and principal pieces of the flagellum [24,129]. The structure of sNHE channels differs from the general structure of Na^+^/H^+^ exchangers because they are constituted by an exchanger domain, a voltage-sensing domain, and a cyclic nucleotide-binding domain [130]. sNHE channels, therefore, are activated by both changes in cAMP levels and voltage. It has been reported that sNHE are the main NHE channels functionally related with K^+^ [8] and Ca^2+^ currents during sperm capacitation [19]. In mice, sNHE could also be coupled to SLC26A3-CFTR complexes; this functional association may result in sustained high HCO_3_^−^ and Cl^−^ levels, which ensure cAMP rises and cytosol alkalinization [81]. Such a functional coupling between SLC26A3 and NHE has also been observed in epithelial cells from intestine [53]. Deficiencies, absence, or dysfunction of sNHE are associated to mouse infertility due to reduced sAC activity and cAMP levels, which lead to impaired hyperactivation of sperm motility [131,132,133,134]. In humans, sNHE also participates in the regulation of sperm motility during capacitation; interestingly, some evidence suggests that this exchanger does not participate in Ca^2+^ influx towards the sperm head or in the acrosomal reaction [129].

Finally, NHE5 has only been identified in the midpiece of mouse sperm [128], and its biological significance in sperm physiology is still unknown. NHA1 and NHA2 transporters are expressed in the principal piece of mouse sperm [8], and both seem to be essential for the activation of sperm motility and cAMP production during ejaculation [120]; yet their specific role during sperm capacitation has not been described.

### 3.2. Voltage-Gated Proton Channel (HVCN1)

The voltage-gated proton channel 1 (HVCN1) belongs to the superfamily of voltage-gated cation channels, which also includes voltage-gated K^+^, Na^+^, and Ca^2+^ channels that differ in structure and gating mechanisms [135,136,137]. Most voltage-gated cation channels are tetramers; each subunit has six transmembrane (TM) segments (S1 to S6), from which S1 to S4 form the voltage-sensing domain (VSD) and S5–S6 form the ion conductance domain [6,138,139]. In contrast, HVCN1 channels are usually found as homodimers, and each subunit has only four TM segments (S1 to S4) that form the VSD and act as a pore (Figure 4) [6,138,139,140]. TM segments are flanked by C- and N-termini, both located in the cytoplasm [6,137,141,142,143,144,145,146]. It has been reported that the gating sensibility of VSD resides in the cytoplasmic region of the S4 segment [139,140,146]. HVCN1 subunits join forming a coiled-coil association that extends intracellularly from each S4 segment, and gate and transport co-operatively [140,146,147,148,149].

HVCN1 channels are highly efficient H^+^ transporters that are not permeable to other ions [138,140]; their activation does not only depend on membrane voltage (Vm), but also on the pH difference (ΔpH = pH_o_ − pH_i_) across the plasma membrane [137,138]. Therefore, when ΔpH = 0, the voltage required for HVCN1 opening is +10 to +30 mV; if ΔpH > 0, which means that the cytosol becomes relatively more acidic than the extracellular medium, the activation requires a potential of around 40 mV/ΔpH unit to more negative voltages. In contrast, if ΔpH < 0, because the cytosol is relatively more alkaline than the extracellular medium, the activation shifts to 40 mV/ΔpH unit to more positive voltages [138,150]. Despite the aforementioned, the mechanisms implicated in the coupling between pH sensing and voltage activation are still unknown. The activity of HVCN1 channels is also modulated by fatty acids, which may affect the gating via the PKC-dependent phosphorylation pathway [6,151,152,153,154]. Moreover, HVCN1 gating and permeability are also highly temperature dependent [140], with the thermosensitivity being placed in the coiled-coil domain [155]. HVCN1 may be inhibited using pharmacological blockers, which bind to the intracellular side of the channel [156], or zinc (Zn^2+^), that joins to the extracellular one [157].

HVCN1 channels have been identified in the plasma membrane of human [158,159], pig [3], and cattle sperm [22], but they are absent from the mouse counterpart [133,137,138]. It has been suggested that in sperm cells, HVCN1 channels export H^+^ more quickly and efficiently than other transporters or exchangers do [133]. In humans, HVCN1 monomers appear as a 32–35 kDa band in immunoblotting assays and are located in the principal piece of the sperm flagellum, close to CatSper [5,159,160]. In cattle, HVCN1 forms a band of around 32 kDa and distributes throughout the principal and terminal pieces of the sperm flagellum [22], whereas in pigs the molecular weight of each subunit is 35 kDa [3,137]. A sperm-specific variant of HVCN1 channels (HVCN1Sper) has been identified in humans and pigs [137]. This variant has a molecular weight of 25 kDa and derives from an alternative splicing at the N-terminus during spermatogenesis by a serine protease [137]. In agreement with this, the expression of HVCN1 and HVCN1Sper is similar between non-capacitated and capacitated sperm cells from the same donor; in contrast, it differs between motile and immotile sperm populations, thus suggesting that they may be relevant for sperm motility [137]. While the activation kinetics are similar between variants, they differ in their pH sensitivity, with the HVCN1Sper channel being less sensitive to pH_i_ changes than the HVCN1 [137,161]. In humans, such an attenuated sensitivity of HVCN1Sper to pH_i_ changes could be essential for the activation of sperm motility after ejaculation [137]. The presence of HVCN1 variants with either or both truncated C- and N- termini has also been described in other tissues [140].

In the epididymal cauda, sperm cells are stored in a quiescent state, which is characterized by ∆pH = 0, with both pH_i_ and pH_o_ being acidic. Activation of sperm motility occurs upon ejaculation thanks to the alkaline seminal plasma. The seminal plasma contains a high concentration of Zn^2+^, which maintains the HVCN1 channels in a close state and prevents premature capacitation (Figure 5) [22,139,140,162]. Zn^2+^ levels, however, decrease progressively throughout the female reproductive tract by dilution in the female fluids, absorption by uterine and oviductal epithelial cells, and albumin chelation [133,140,158,163]. Moreover, during the sperm journey throughout the female tract, the pH_o_ becomes progressively more alkaline [140,160]. Altogether, the progressive decrease of extracellular Zn^2+^ levels and progressive increase of pH_o_ induce HVCN1 gating and H^+^ extrusion [40]; moreover, the high HVCN1 activity in capacitated sperm could be also related to the phosphorylation of their subunits, presumably by PKC [11,134]. In capacitated sperm of humans and cattle, HVCN1 induces a fast alkalinization at the principal piece, which is crucial for the switch in motility pattern as it drives the activation of CatSper channels and pH-dependent proteins of the axoneme [11,22,133,137,159,161]. In both species, HVCN1 channels are also required for acrosomal exocytosis [22,161], although some studies performed in human sperm support that the alkalinization of the sperm head does not only rely upon HVCN1 but also on bicarbonate transporters [11]. Interestingly, in pig sperm, HVCN1 is essential for motility hyperactivation during in vitro capacitation but not for progesterone-induced acrosomal exocytosis; moreover, in this species, such a close relationship between HVCN1 activity and Ca^2+^ influx has not been observed [3].

HVCN1 channels may also be activated by albumin, which is highly abundant in the uterine fluid (500 μM) as compared to the seminal fluid (15 μM) [161,164]. It has been reported that one albumin molecule can bind external HVCN1 residues linking S3 and S4 transmembrane segments of the two VSDs of one dimer, thus favoring channel opening [136,161]. Interestingly, while albumin has a high specificity for HVCN1 channels, it does not have this high affinity for other voltage-gated channels; in the absence of albumin, HVCN1 activates slowly and shows fast deactivation, whereas albumin-HVCN1 complexes are characterized by fast activation, slow deactivation, and high current magnitude [161,165]. Such a regulatory effect has been reported to be similar between HVCN1 and HVCN1Sper variants [161]. In a recent study, Zhao et al. [161] demonstrated that the incubation of non-capacitated human sperm in the presence of albumin resulted in an increased H^+^ current efflux across HVCN1 channels, pH_i_ alkalinization, and sustained Ca^2+^ influx through CatSper; even the addition of progesterone to the medium induced the acrosomal exocytosis. Finally, it is worth bearing in mind that albumin can also act as a cholesterol acceptor during sperm capacitation [14,166].

## 4. Conclusions

Alkalinization of sperm cytosol during capacitation is conducted by HCO_3_^−^ influx and H^+^ efflux, and it is characterized by a fast HCO_3_^−^ uptake that triggers sperm capacitation by activating the cAMP/PKA pathway, followed by a sustained HCO_3_^−^ entrance and H^+^ output that are essential for the hyperactivation of sperm motility, membrane hyperpolarization, and acrosomal exocytosis. Studies on HCO_3_^−^ uptake during sperm capacitation have been mainly conducted in rodents and humans. In these species, while the initial HCO_3_^−^ entrance is mediated by Na^+^/HCO_3_^−^ cotransporters (NBC) belonging to the SLC4 family, the NBC types implicated in this transport and their localization and activation are still unknown. Sustained HCO_3_^−^ entrance requires different Cl^−^/HCO_3_^−^ exchangers pertaining to SLC4 and SLC26 families; their gating is regulated by phosphorylation through the PKA pathway, which is activated by NBC, and it is essential for the maintenance of this pathway during the sperm capacitation. To date, five different anion exchangers have been identified in the plasma membrane of rodents and humans, SLC4A1 and SLC4A2, and SLC26A3, SLC26A6, and SLC26A8, which are mainly located in the equatorial region of the sperm head and the midpiece. During sperm capacitation, these exchangers induce a moderate increase of pH_i_, and are essential for the regulation of sperm movement and for triggering the acrosomal reaction via interaction scaffolding proteins and other transporters. Interestingly, the activity of SLC26 exchangers is closely related to Cl^−^ inward, mainly through CFTR but also other Cl^−^ transporters; this coupled HCO_3_^−^ and Cl^−^ entrance to the sperm cell seems to be essential for membrane hyperpolarization. Unfortunately, little data on the expression, distribution, and regulation of SLC4 and SLC26 transporters in the sperm plasma membrane of other mammal species exist.

Proton extrusion induces a fast alkalinization in capacitated sperm cells, which is necessary for motility hyperactivation and membrane hyperpolarization. To date, two different proton transporters have been reported to take part in the increase of pH_i_: NHE and HVCN1; yet differences in the types, localization, regulation, and relevance of these channels during sperm capacitation exist between species. In rodents, H^+^ transport largely relies on NHE channels as these species do not express HVCN1. In mice, the plasma membrane of mid- and principal pieces contains five types of NHE, NHE1, NHE5, sNHE, NHA1, and NHA2. The biological relevance of these different exchangers is yet to be elucidated; only the activation of sNHE has been correlated to increased Ca^2+^ and K^+^ transport across the plasma membrane, two events known to be involved in the hyperactivation of sperm motility and membrane hyperpolarization, respectively. In humans, cattle, and pigs, HVCN1 is essential for the alkalinization of the sperm flagellum and further hyperactivation of sperm motility through a Ca^2+^-dependent pathway. While in these species, NHE channels are also present in the sperm plasma membrane, their types and physiological roles are still poorly studied.

Scientific evidence suggests that the activation of these bicarbonate and proton channels occurs after ejaculation due to differences in the composition between the seminal and female fluids. Further studies must be focused on the physiological relevance of female hormones (estrogen and progesterone) on the activation of these channels after ejaculation.

## Figures and Tables

**Figure 1 ijms-23-06333-f001:**
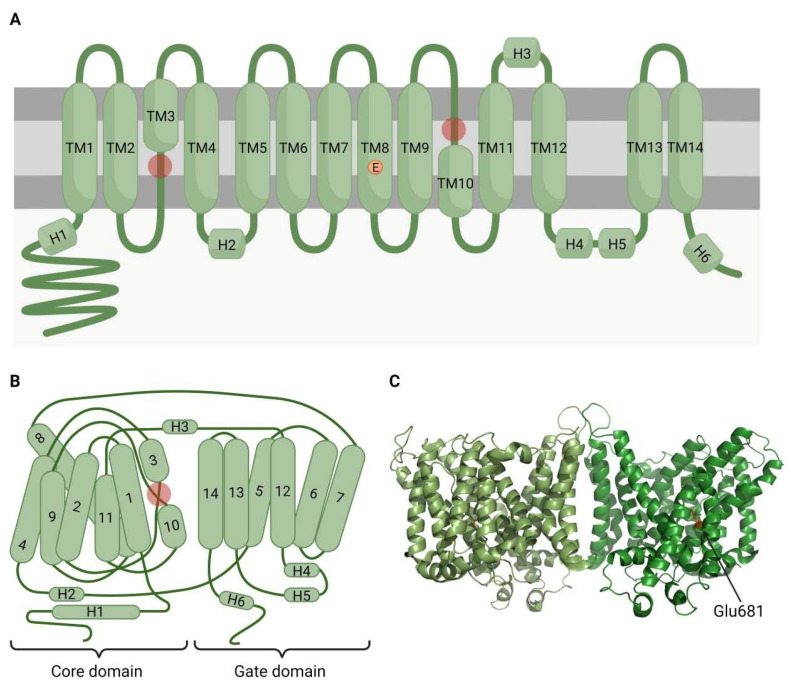
Structural characteristics of SLC4 channels. Bicarbonate transporters from the SLC4 family have slightly different functions, but the structure is generally conserved between them. The structure presented in this figure corresponds to SLC4A1 channel (PDB reference 4YZF). (**A**) SLC4 channels present 14 transmembrane α-helices (TM1–14) connected through loops that occasionally present amphipathic helices (H1–6). TM3 and TM10 are two half-helices, and their N-termini regions form the active site (red circles). In the TM8 segment, Glu681 (orange, E) is a blocker of the active site. (**B**) The folded protein conforms two different domains: the core domain, which is formed by helices TM1–4 and TM8–11; and the gate domain, which consists of TM5–7 and TM12–14. N-termini of TM3 and TM10 face each other to form the active site (red circle). (**C**) SLC4 channels form dimers that interact through a dimerization interface constituted by TM6 and TM7, which are part of the gate domain.

**Figure 2 ijms-23-06333-f002:**
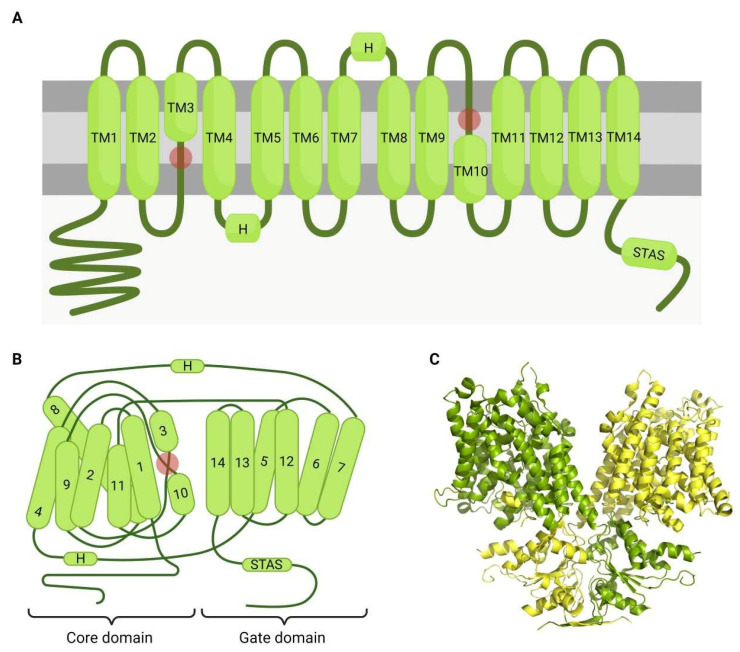
Structural characteristics of SLC26 channels. Bicarbonate exchangers from the SLC26 family present a highly similar structure to SLC4 channels. The structure presented in this figure corresponds to the SLC26A9 channel (PDB reference 7CH1). (**A**) SLC26 channels present 14 transmembrane α-helices (TM1–14) connected through loops that occasionally present amphipathic helices (H). TM3 and TM10 are two half-helices, and their N-termini regions form the active site (red circles). At the C-terminal region, a sulphate transporter anti-sigma factor antagonist domain (STAS), which is involved in SLC26 trafficking, and protein–protein interaction and regulation, which includes the interaction with the cystic fibrosis transmembrane conductance regulator channel (CFTR). (**B**) The folded protein conforms two different domains: the core domain, which is formed by helices TM1–4 and TM8–11; and the gate domain, which consists of TM5–7 and TM12–14. The N-termini of TM3 and TM10 face each other to form the active site (red circle). (**C**) SLC26 channels form dimers, with the STAS domain being excluded from the dimerization domain, since it must be free to interact with other proteins.

**Figure 3 ijms-23-06333-f003:**
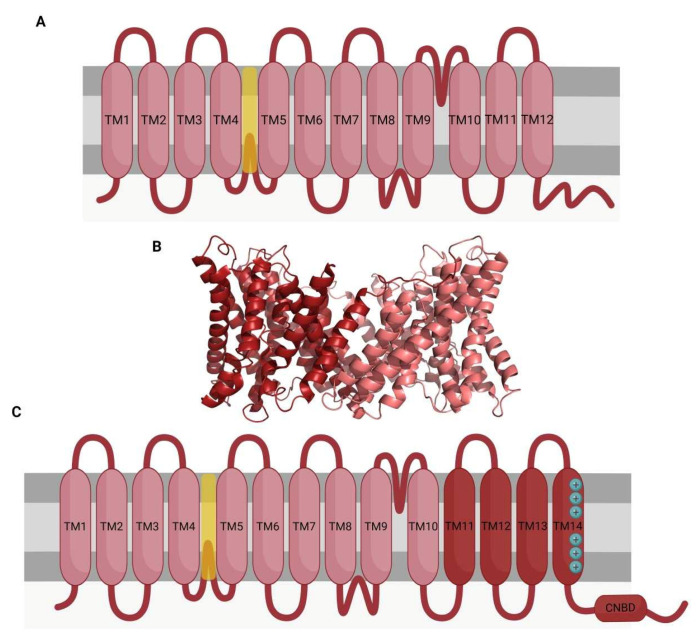
Structural characteristics of NHE channels. Sodium–proton exchangers from the SLC9 family of channels present slightly variable structures, with the sNHE channel presenting some characteristic features. (**A**) SLC9 channels generally present 12 transmembrane α-helices (TM1–12) connected through loops. An ion permeation pathway is formed between TM4 and TM5 (yellow). The C-terminal domain presents multiples sites of interaction with other proteins. (**B**) Quaternary conformation of SLC9 channels, which tend to form dimers, acquire a more stable conformation. The structure presented in this figure corresponds to the SLC9A1 channel (PDB reference 7DSW). (**C**) The sNHE channel structure is different from other NHE channels, mainly for TM11–14. This domain resembles the voltage-sensing domain of other channels. TM14 has a high number of basic residues, which corresponds to the voltage-sensing region (blue, +). The C-terminal region contains, among the sites of interaction with other molecules, a cyclic nucleotide-binding domain (CNBD).

**Figure 4 ijms-23-06333-f004:**
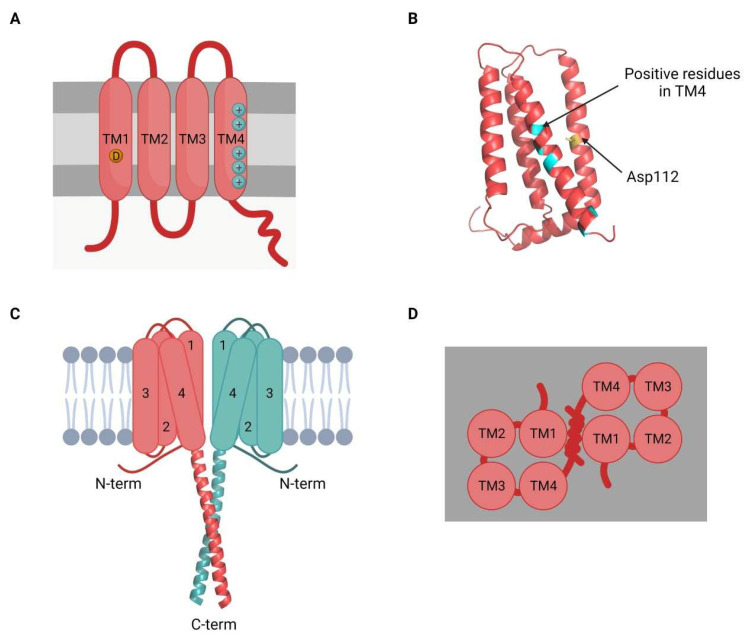
Structural characteristics of the voltage-gated proton channel (HVCN1). (**A**) The proton channel HVCN1 presents four transmembrane α-helices (TM1–4) connected through three loops. In TM1, the residue Asp112 is highly conserved and is critical for proton selectivity (yellow, D). The TM4 segment presents multiple basic residues, which correspond to the voltage-sensing region (blue, +). Both the loop between TM1 and TM2, and the C-terminal coiled coil are essential for dimerization. (**B**) Each monomer folds to form a single ion permeation pathway that also contains the voltage-sensing domain (VSD) (PDB reference 5OQK, lacking the coiled-coil domain located at the C-terminal region). (**C**) The quaternary structure of HVCN1 consists of the formation of dimers, and the two coiled coils interact intracellularly (PDB reference 3VMX). (**D**) Dimers present two different permeation pathways, each corresponding to a different monomer.

**Figure 5 ijms-23-06333-f005:**
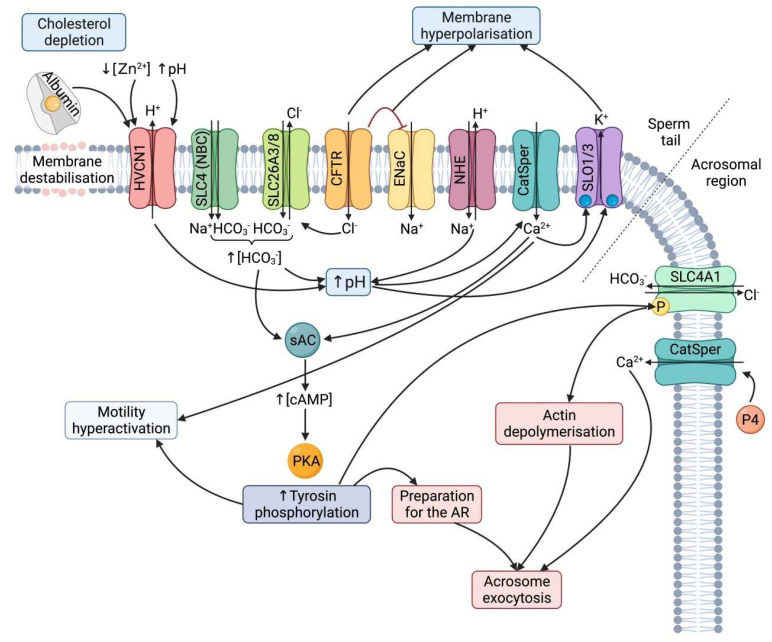
Membrane channels in sperm capacitation and the acrosome reaction. When sperm enter the female tract, the plasma membrane is partially destabilized because of cholesterol depletion. Albumin does not only contribute to cholesterol depletion, but also activates HVCN1. This channel is also activated by the higher pH in the female tract and the lower concentration of Zn^2+^ compared to the male tract. One of the first events in the capacitation pathway is the activation of SLC4 (NBC) channels that contribute to the increase in bicarbonate concentration. This increase is also achieved through the cooperation of SLC26 channels with CFTR. The activation of CFTR inhibits ENaC channels, and these two events contribute to plasma membrane hyperpolarization. Intracellular pH alkalinization occurs in response to both HCO_3_^−^ influx and H^+^ efflux through the plasma membrane, which activates both CatSper and SLO channels. Both high pH and increased concentration of Ca^2+^ activate the soluble adenylate cyclase (sAC), which increases cAMP levels and, in turn, PKA activity. The increase in tyrosine phosphorylation prepares the sperm for the AR, and hyperactivates sperm motility. In addition, downstream of the signaling pathways of tyrosine phosphorylation, SLC4A1 channels in the acrosome region are phosphorylated, which activates actin depolymerization in this region. The increase in progesterone levels that sperm encounter in the oocyte vicinity activates CatSper channels in the sperm head, which trigger the acrosome reaction.

## Data Availability

Not applicable.

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
