# Peer review of "A Review on the Role of Bicarbonate and Proton Transporters during Sperm Capacitation in Mammals"

_ijms, 2022, doi:10.3390/ijms23116333_

Round 1

Reviewer 1 Report

Dear author,

The manuscript entitled "A review on the role of bicarbonate and proton transporters during sperm capacitation in mammals" is a review which gathers the latest knowledge on the structure, localization, physiological role, and regulation mechanisms of ion transporters implicated in cell alkalinization during sperm capacitation in mammals, including bicarbonate transporters (Na+/HCO3- cotransporters and Cl-/HCO3- exchangers) and proton transporters (NHE and HVCN1 channels). The plasma membrane of sperm cells contains different ion channels implicated in the increase of inner (pHi) by favouring either bicarbonate entrance or proton efflux. Alkalinization of sperm cytosol during capacitation is conducted by HCO3- influx and H+ efflux, and it characterizes by a fast HCO3- uptake that triggers sperm capacitation by activating the cAMP/PKA pathway, followed by a sustained HCO3- entrance and H+ output that are essential for the hyperactivation of sperm motility, membrane hyperpolarization, and acrosomal exocytosis. Proton extrusion induces a fast alkalinization in capacitated sperm cells, which is necessary for motility hyperactivation and membrane hyperpolarization.

 The manuscript can be accepted for publication after minor revision  .

Bellow you will find very little corrections that you have to do in  your manuscript

 P16 L537: and is characterized instead of  and it characterizes

Author Response

Reviewer's comment: P16 L537: and is characterized instead of  and it characterizes

Answer: Thank you very much for your comment. We have reviewed and corrected this sentence in the reviewed versión. The correction has been highlighted in red.

Reviewer 2 Report

This study is prepared in a very interesting and interesting way. It fully covers the role of bicarbonates and protein transporters on the functional parameters of mammalian sperm. Schematic drawings enrich these works enormously. I have no merit or linguistic remarks.

1.There is only thing I did not find was the information about the dynamics of the protein channels, are they constantly in their number and position or does it change during the development of the sperm and the transition to different phases of activity during fertilization? Is there an infuence of hormones/hormonally active chemicals in sperm plasma on these channels? Plaese add information to the ms text.

Author Response

Reviewer’ comment: This study is prepared in a very interesting and interesting way. It fully covers the role of bicarbonates and protein transporters on the functional parameters of mammalian sperm. Schematic drawings enrich these works enormously. I have no merit or linguistic remarks.

Answer: We sincerely appreciate your comment.

Reviewer’s comment: There is only thing I did not find was the information about the dynamics of the protein channels, are they constantly in their number and position or does it change during the development of the sperm and the transition to different phases of activity during fertilization? Is there an infuence of hormones/hormonally active chemicals in sperm plasma on these channels? Plaese add information to the ms text.

Answer: Please note that the review is focused on the sperm capacitation, so we sincerely believe that, despite being of great interest, the addition of information related to the changes in number and position of these channels during sperm development and fertilization does not agree with the scope of the review.

Moreover, as extensively reported in the review, the activation of bicarbonate and proton channels occurs during sperm transit throughout the female tract, as a result of differences in the composition between seminal fluid and female fluids. Nevertheless, we consider your suggestion very interesting, and so we performed a deep bibliographic revision of this topic. Several evidences demonstrate that progesterone has a relevant role in triggering acrosome reaction and hyperpolarization by acting on CatSper channels; this is clearly indicated in the Figure 5 of the review. As far as the authors concern, no data exist on the regulation of proton or bicarbonate channels by progesterone or other female hormones. However, due to the relevance of your comment at the end of the Conclusion section we have added the sentence “Scientific evidence suggests that the activation of these bicarbonate and protons channels occurs after ejaculation due to differences in the composition between the seminal and female fluids. Further studies must be focused on the physiological relevance of female hormones (estrogen and progesterone) on the activation of these channels after ejaculation”. This change has been highlighted on the text.